# Linking Career-Related Social Support to Job Search Behavior Among College Students: A Moderated Mediation Model

**DOI:** 10.3390/bs15030260

**Published:** 2025-02-24

**Authors:** Zhangbo Xiong, Meihong Zeng, Yi Xu, Bin Gao, Quanwei Shen

**Affiliations:** 1School of Education, Shanghai Normal University, Shanghai 200234, China; 2School of Psychology, Shanghai Normal University, Shanghai 200234, China

**Keywords:** career-related social support, career decision-making self-efficacy, employment pressure, job search behavior, college students

## Abstract

Background: Career-related social support (CRSS) significantly influences job search behavior (JSB). However, the mechanisms and boundary conditions through which CRSS impacts JSB, particularly during the COVID-19 pandemic, remain unclear. This study examines the mechanisms and boundary conditions through which CRSS influences the JSB among final-year college students during the COVID-19 pandemic, guided by social cognitive career theory (SCCT). Methods: A cross-sectional online survey was conducted in 2021 among 596 final-year college students from two public universities in China, during the COVID-19 pandemic. The survey collected data on CRSS, CDMS, employment pressure, and JSB. Results: The findings revealed that career decision-making self-efficacy (CDMS) partially mediates the relationship between CRSS and JSB. Moreover, employment pressure moderates both the direct and mediated pathways; when employment pressure is low, CRSS does not significantly enhance JSB, whereas under a high employment pressure, JSB significantly increases, with higher CRSS. In the mediated pathway, higher CDMS strongly predicts JSB under low employment pressure, but its influence diminishes as the employment pressure rises. Conclusions: These findings underscore the critical roles of CDMS and employment pressure in shaping JSB, offering valuable insights for career support initiatives to facilitate the transition of graduates into the workforce during challenging periods.

## 1. Introduction

The issue of youth employment has always been a hot topic of concern in the society, especially in China, where job search behavior (JSB) and successful employment of college graduates have garnered widespread attention from government agencies, universities, and families. Data show that the number of Chinese college graduates reached 11.58 million in 2023, and the Chinese government has placed a stronger emphasis on “stabilizing employment” ([42]). As of June 2023, the Chinese government reported a historically high unemployment rate of 21.3% for young Chinese workers aged 16 to 24 ([43]). Importantly, the individuals’ JSB has been found to be closely related to their employment success, and thus has received extensive attention from numerous scholars ([9]; [11]; [25]). Positive JSB not only helps individuals secure more interviews and job opportunities, facilitating their successful employment ([24]; [49]; [11]), but it is also an effective predictor of successful reemployment of the unemployed ([27]; [5]). The COVID-19 pandemic has severely disrupted the world economy, making the employment situation for young people even more challenging, particularly in China, where the employment pressure on college graduates has significantly increased ([31]). Therefore, it is imperative to further investigate the various factors affecting the successful job search and employment in the post-COVID-19 era.

Numerous research findings have shown that obtaining social support and career mentoring can benefit those who are encountering challenges (e.g., during an economic recession) while searching for employment ([47]; [26]; [7]). However, studies examining the correlation between career-related social support (CRSS) and JSB have yielded inconsistent results. Specifically, while most studies indicate that social support is a facilitating force for JSB ([59]; [47]; [12]), a recent investigation found that individuals with increased social support were less inclined to be involved in JSB ([28]). Therefore, there may be potential moderating variables that can affect the degree and direction of the link between social support and JSB.

Several psychological frameworks, such as the theory of planned behavior ([53]; [34]), the expectancy–value theory ([39]), and the self-regulation theory ([58]) have been widely used to study JSB. However, the social cognitive career theory (SCCT) is particularly suitable for this study as it emphasizes the dynamic interplay between cognitive, affective, and environmental factors in career-relevant activities ([29]). According to SCCT, an individual’s JSB is influenced by a combination of social support, occupational self-efficacy, and employment-environment factors ([74]; [12]; [64]). Thus, this research, set against the backdrop of the COVID-19 pandemic, explores the joint impact of CRSS, career decision-making self-efficacy (CDMS), and employment pressure on JSBs among recent college graduates, based on the SCCT theoretical framework, which in turn provides empirical evidence and inspiration to intervene in university students’ JSBs.

Social support is domain-specific, and the different domains of social support may have different impacts on an individual’s JSB. Considering that earlier studies conceptualized social support in a broad sense ([59]; [47]; [12]), little is known about the association between CRSS and JSB. In this paper, the term CRSS is used to define the social support that individuals receive from the four different sources of social resources (material support, advice support, emotional support, and information support)—their parents, relatives, friends, and teachers in career-related domains ([20]), while the term “social support” in a broader sense is usually used to also include the social resources that people perceive as available or that are actually provided in the general context ([14]). According to the SCCT, an individual’s CRSS can influence his or her career choices and development ([29]). College students during the job search process typically receive support and assistance from family members (e.g., parents), schools (e.g., instructors), and the government (employment policies). From the social network perspective, research has found that family members can provide the individuals with career-related information and job search advice, which enhances individuals’ JSBs and, consequently, increases their likelihood of securing interviews ([9]). Cross-sectional studies indicated that CRSS was effective in promoting vocational identity and career adaptability in Chinese school students ([68]; [67]). Longitudinal evidence demonstrated that CRSS contributed to enhancing future perspectives, career maturity, and career adaptability in Chinese adolescents and undergraduates ([6]; [33]; [16]). A cross-sectional study focusing on new entrant job seekers found that several types of support messages (e.g., informational, emotional, and companionship support) were positively associated with JSB ([47]). An 8-month survey of final-year college students found that mentor career support stimulated job search intentions and career planning behavior, and also reduced self-defeating JSBs ([46]). According to a longitudinal study by [25] ([25]), mentors can provide social and psychological support to young adults, such as advice, which can help them develop a clear sense of their future work selves. This, in turn, enhances their JSBs. Higher levels of CRSS can enhance the career adaptability and employability of college graduates ([63]; [62]), consequently promoting their JSBs. Based on the theoretical perspective of the SCCT and the above empirical evidence, CRSS may serve as a positive factor driving individuals’ JSBs. Therefore, this study hypothesizes that CRSS positively predicts JSB (H1).

Career decision-making self-efficacy (CDMS) is an individual’s subjective confidence in one’s ability to successfully accomplish vocational choice-related tasks ([56]). According to SCCT, CRSS serves as a primary source of information for career self-efficacy ([23]). Empirical research indicated that both peer social support and perceived parental support contribute to enhancing an individual’s CDMS ([8]; [15]). A cross-sectional study shows that CRSS can directly influence Chinese college students CDMS ([60]). Similarly, Indonesian college students with more peer support (e.g., peer role models, career information, and advice) exhibit higher CDMS ([1]). On the other hand, CDMS serves as a promoting factor for JSBs among college students ([69]). Specifically, CDMS can enhance an individual’s job search self-efficacy ([36]), increase employment search willingness and frequency, thereby promoting JSB ([4]; [65]; [74]). Numerous studies have also found that CDMS significantly and positively predicts the individuals’ JSBs ([49]; [13]). Therefore, this study hypothesizes that CDMS mediates the link between CRSS and JSB (H2).

The SCCT posits that the individuals’ perceived external environment (e.g., social support, employment pressure) can interact with their cognitive characteristics (e.g., CDMS) to jointly influence their career development ([30]). In other words, while CRSS may predict JSB through CDMS, its impact may vary among different individuals. That is, there may be individual differences in the association between CRSS and JSB among undergraduates. Employment pressure refers to the stress response that the individuals may experience when searching for, or choosing a job, due to internal or external factors ([38]). Additionally, an empirical investigation suggested that during the COVID-19 epidemic, university students experienced a reduction in job availability, and faced stronger employment stress from the job market ([64]; [31]). In fact, employment pressure can have double-edged effects on the individuals’ career-related outcomes. Research has shown that university students with high employment pressure tended to have higher employment anxiety during the COVID-19 ([48]), poorer levels of mental health ([21]) and lower occupational delay of gratification ([54]). Furthermore, higher employment pressure can reduce an individual’s CDMS ([19]) and job-searching self-efficacy ([64]). On the contrary, unemployed people with a high perceived social pressure to apply for a job, tended to be more active in their search and actually found a paying job ([5]). Importantly, the emergence of the COVID-19 outbreak led to an urgent rise in JSB, and this effect continued into the post-pandemic period ([40]). Furthermore, recent research suggested that employment pressure could serve as a moderating variable between individual factors (e.g., psychological capital, proactive personality) and adaptive outcomes (e.g., CDMS, depression) ([21]; [71]; [19]). However, limited research has tested its moderating effect between the CRSS and JSB in the context of COVID-19. Thus, this study proposes that employment pressure can moderate the direct and mediated pathways through which CRSS influences JSB (H3).

In summary, drawing on the SCCT, the current research constructs a research model (see Figure 1) to analyze “how” and “when” CRSS influences JSBs among graduating college students. The main objectives of the research are threefold: (1) to test whether CRSS can directly predict final-year college students’ JSBs; (2) to investigate whether CDMS mediates the relationship between CRSS and the graduating college students’ JSBs; (3) to explore whether employment pressure moderates the direct and mediated paths through which CRSS affects JSB.

## 2. Materials and Methods

### 2.1. Participants and Procedure

Convenient cluster sampling was employed to select the participants for an online questionnaire survey among final-year college students from two universities in the central region of China. A total of 680 questionnaires were distributed by class units, and 596 valid participants were collected after eliminating invalid responses. The recovery rate for valid samples was 87.6%. The respondents had a mean age of 21.74 years (SD = 1.40), with 166 males (27.9%) and 430 females (72.1%). Among the participants, 153 came from urban areas (25.7%), while 443 came from rural areas.

The survey was carried out during COVID-19, in the period between May and June 2021. This timeframe coincided with the graduation season for Chinese university students, during which final-year college students actively sought employment. Response to the survey was completely voluntary, and informed consent was obtained from all the respondents before completing the electronic questionnaire. This research was granted ethical approval by the corresponding author’s institutional ethics committee (protocol code SHNU-IRB-2020096).

### 2.2. Measures

Career-related social support scale: The CRSS scale, originally devised by [20] ([20]), comprises 20 items categorized into four dimensions: material support, advisory support, emotional support and informational support. The scale employs a 5-point Likert scale (1 represents “almost none”, 5 represents “a lot”), with higher scores indicating greater levels of CRSS. The alpha coefficient of this scale was 0.91.

Career decision-making self-efficacy scale: The CDMS scale, originally devised by [56] ([56]) and revised by [44] ([44]), consists of 39 items across five dimensions: self-appraisal, information gathering, goal setting, plan making, and problem solving. It utilizes a 5-point Likert scale, where 1 indicates “almost no confidence” and 5 indicates “completely confident”. Higher scores reflect a higher level of CDMS. The alpha coefficient of this scale was 0.90.

Employment pressure scale: The employment pressure scale, initially devised by [37] ([37]), comprises a total of 14 items across four dimensions: personal factors, family environment, school background, and professional experience. It employs a 5-point Likert scale, ranging from “1 = almost no pressure” to “5 = very stressful”. Higher score indicates a greater employment pressure. The alpha coefficient of this scale was 0.94.

Job search behavior scale: The job search behavior scale, initially devised by [3] ([3]), consists of 12 items encompassing two dimensions: job search preparation (e.g., searching for career information, setting career goals, and formulating action plans) and job search action (e.g., submitting resumes and attending interviews). It employs a 5-point Likert scale (1 representing “0–1 times”, 5 representing “five times or more”). Higher score indicates a higher level of JSB. The Chinese version of this scale has demonstrated good reliability and validity in a previous study ([65]). The alpha coefficient of this scale in this study was 0.94.

### 2.3. Statistical Analyses

To ensure the quality of our data, we conducted data cleaning. Specifically, we eliminated the invalid subjects, which resulted in 596 remaining participants. Additionally, no outliers were detected in the analyzed dataset. Then, we conducted Pearson’s correlation analyses between the major variables using SPSS 24.0, and then used the PROCESS macro 4.0 to examine the research hypotheses model. Prior research has shown that individual characteristics such as gender, Hukou (urban or rural), and family socioeconomic status (SES) may influence an individual’s JSB ([69]; [32]). Therefore, in the regression analysis, these three factors were considered as control variables. To measure the SES, we used a single item ([73]), which allowed respondents to identify their social status from 1 (lower class) to 5 (upper class). As this study employs a questionnaire survey method, there is a possibility of common method bias (CMB). To address this concern, unrotated principal component factor analysis was performed in the initial analysis. The results of this analysis revealed a total of eleven factors with eigenvalues exceeding one. The variance of the first factor is 27.37%, which is below the criterion of 40% ([55]). Therefore, it can be concluded that there is no significant problem with the CMB in this investigation.

## 3. Results

### 3.1. Descriptive Statistics and Correlation Analysis

The correlation analysis, as shown in Table 1, indicated that CRSS, CDMS, and JSB are all significantly positively correlated with each other. The employment pressure was negatively linked to CDMS, and on the contrary, was positively linked to JSB.

### 3.2. Testing the Mediation Model

Using the bias-corrected percentile Bootstrap method (Bootstrap = 5000) and Model 4 from the Process macro, the mediation path of CRSS → CDMS → JSB was examined. The results (see Figure 2) indicate that CRSS not only positively predicts JSB (β = 0.16, *p* < 0.001), but also positively predicts CDMS (β = 0.45, *p* < 0.001). Similarly, CDMS significantly and positively predicts JSB (β = 0.27, *p* < 0.001). The Bootstrap 95% CI for the mediation effect is [0.07, 0.19], which does not contain 0. Therefore, CDMS partially mediates the link between CRSS and JSB, with the mediation effect (0.12) accounting for 41.38% of the total effect (0.29). Thus, H1 and H2 are supported.

### 3.3. Testing the Moderated Mediation Model

Using the Process macro (Model 59) to test the hypothetical model, the results (as shown in Table 2) reveal that the interaction effect of CRSS and employment pressure significantly predicts JSB (β = 0.11, *p* < 0.01). The Bootstrap 95% CI for the interaction term is [0.04, 0.18], which does not include zero, indicating that employment pressure positively moderates the link between CRSS and JSB. The interaction effect of CRSS and employment pressure does not significantly predict CDMS (β = 0.01, *p* > 0.05), meaning that employment pressure does not moderate the link between CRSS and CDMS. The interaction effect of CDMS and employment pressure significantly predicts JSB (β = −0.07, *p* < 0.05). The Bootstrap 95% CI for the interaction term is [−0.13, −0.01], which does not include zero. This implies that employment pressure negatively moderates the link between CDMS and JSB. Therefore, H3 is partially supported.

To further elucidate the moderation effect described above, the employment pressure was categorized into high (M + 1SD) and low groups (M − 1SD), and simple slope analyses were carried out. The results (see Figure 3) reveal that when the employment pressure is low, there is no significant upward trend in JSB with increasing CRSS (β = 0.06, *p* > 0.05), suggesting that for college students under relatively low levels of stress or concern about employment, CRSS does not have a substantial impact on their engagement in job search activities. However, when the employment pressure is high, there is a significant upward trend in JSB with increasing CRSS (β = 0.25, *p* < 0.001). This indicates that, under high employment pressure, college students are more responsive to career-related social support, with increased support leading to a marked increase in JSB.

Similarly, as depicted in Figure 4, when the employment pressure is low, there is a significant upward trend in JSB with increasing CDMS (β = 0.38, *p* < 0.001). This suggests that in low-pressure environments, college students are more likely to engage in job search activities. However, when employment pressure is high, the upward trend in JSB with increasing CDMS is less pronounced (β = 0.25, *p* < 0.001). This weaker relationship indicates that, under higher levels of employment pressure, college students may experience diminished confidence in their job search efforts, which can hinder the positive effects of CDMS.

## 4. Discussion

This study, grounded in the SCCT, examined the mechanisms and boundary conditions through which CRSS influences the JSBs in college students. Specifically, the research highlights the critical role of CRSS in shaping JSB, particularly during challenging times, such as the COVID-19 pandemic, or during periods of high employment pressure. By examining how CRSS not only directly impacts job search efforts but also acts as a buffer in times of stress, this study provides valuable insights into the ways social support can mitigate the negative effects of employment pressure on the students’ career development. Furthermore, this research identifies key boundary conditions, such as the moderating effect of employment pressure, that influence the efficacy of CRSS in promoting job search behavior. These findings underscore the importance of considering external contextual factors when designing interventions for student support. In response to the existing research gaps, the study proposes the design of tailored mentoring programs that integrate CRSS and employment pressure management strategies. Such programs can provide personalized support, enhancing the students’ coping mechanisms and job search strategies. Finally, the findings have important implications for public policy. Policymakers can use this research to inform the development of policies that create supportive environments for college graduates, from academia to the workforce.

The results confirm that CRSS positively predicts JSBs in college students, which was in line with the prior related studies ([52]; [63]). Additionally, these results further support SCCT, highlighting the significance of CRSS as an essential environmental factor influencing JSBs among college students. Furthermore, the conservation of resources (COR) theory, as introduced by [17] ([17]), offers further insight into these research findings. Specifically, individuals are encouraged to preserve and manage their resources, continuously striving to maintain and increase the resource base that contributes to achieving goals and desired outcomes. According to the COR theory, CRSS serves as a critical social resource that helps the individuals safeguard their existing resources and acquire new ones ([51]). In other words, individuals with higher levels of CRSS possess valuable social network resources that provide them with more job search information and employment opportunities. They are also more likely to receive material and emotional support from external sources, enabling them to be involved in more employment search attempts and activities ([9]; [46]). Conversely, for individuals with decreased CRSS, their social resources are limited, making it challenging for them to access ample job search information and receive external guidance and assistance. Consequently, their JSBs are more limited due to constraints on their social resources. Therefore, enhancing CRSS among college students is an important approach to promote their proactive JSBs.

This study confirmed that CDMS plays a partial mediating role in the link between CRSS and JSB. This suggests that CRSS acts as a facilitating factor for CDMS, indicating that higher levels of CRSS contribute to enhancing the CDMS of college students. This finding further supports the SCCT as different sources of social support are among the significant sources of individual CDMS. Empirical research has shown that diverse sources of social support (e.g., family support, peer support, and teacher support) can provide individuals with career-related advice and guidance ([1]). This, in turn, helps enhance the career exploration among students ([57]) and their career adaptability ([61]), overcoming hesitancy in career choices ([22]), increasing career certainty ([2]), and boosting CDMS. Furthermore, this study finds that CDMS significantly enhances JSBs, aligning with previous research ([74]; [50]). This is due to the fact that individuals with a high CDMS are more active in managing career-related work, learning, and personal life ([60]), have stronger career planning abilities ([72]), and engage in more career exploration ([45]), thus exhibiting more JSB ([18]).

This study, for the first time, confirmed that employment pressure can moderate the direct pathway of CRSS affecting JSB as well as the second stage of the mediating pathway. This result further supports the SCCT ([30]), which suggests that individual factors (e.g., CDMS) and external environmental factors (e.g., CRSS and employment pressure) interact to jointly influence the JSBs of college students. On one hand, employment pressure negatively moderates the link between CDMS and JSB. Specifically, employment pressure weakens the positive impact of CDMS on JSB. The reason behind this is that higher levels of employment pressure may reduce an individual’s self-efficacy and confidence in successful employment ([66]; [70]), subsequently diminishing their JSB. For example, a graduate facing heightened employment pressure in a saturated job market may struggle to maintain motivation and resilience during the job search process. In such cases, tailored CRSS, such as personalized career counseling, skill-building workshops, or networking opportunities, can provide essential resources to manage these challenges. This support can help students regain their sense of control and optimism, mitigating the detrimental effects of employment pressure.

On the other hand, employment pressure positively moderates the link between CRSS and JSB. That is, when external employment pressure is high, higher levels of CRSS can enhance individual JSB. Conversely, when external employment pressure is low, even lower levels of CRSS do not significantly reduce individual JSB. Therefore, in the current context of a sluggish global economy and increased employment pressure, these findings are expected to provide valuable insights for interventions aimed at enhancing JSB among university students. In addition, this result can be explained by the theory of optimal matching, which posits that some kinds of social support are optimally matched to specific stressors by facilitating appropriate coping responses, and that effective social support needs to be tailored to the recipient’s needs ([10]). Although supportive behaviors are typically intended to provide assistance, they can sometimes have counterproductive effects on the individuals receiving help ([35]). In other words, when individuals perceive reduced employment pressure, they are more inclined to believe they have the capability to handle external pressures autonomously, and may not require external job-related support to participate in employment seeking activities. At such times, the provision of high levels of social support may potentially inhibit individual JSB, because when externally provided support mismatches with personal needs, it could threaten self-esteem ([52]) and autonomy ([41]), thus weakening the positive predictive effects of social support on JSB. The discussions above provide insights suggesting that CRSS should take into account the current job market environment and an individual’s actual needs.

### 4.1. Implications

The results of this study hold significant theoretical and practical implications. Firstly, to our best knowledge, this study is the first to integrate CRSS and employment pressure within the framework of SCCT, thereby expanding and enriching the scope of research in the field of JSB. Secondly, this research investigates the indirect effect of CDMS on the link between CRSS and JSB in college students, providing a clearer understanding of the mediating mechanism. Additionally, by introducing employment pressure as a moderating variable based on SCCT, this study further clarifies the boundary conditions influencing the impact of CRSS on JSB. Moreover, the findings offer valuable insights into practical interventions aimed at improving college students’ JSBs. CRSS and CDMS play crucial roles in promoting JSBs among college students, CRSS as an external factor and CDMS as an internal factor. Notably, the effectiveness of CRSS needs to be matched accordingly to the psychological needs of job seekers. When job seekers are experiencing high employment pressure, indicating a lack of effective coping mechanisms to deal with external job market pressures, the provision of career support can be particularly helpful, enhancing job search confidence and aiding in overcoming the employment pressure. In light of the economic challenges posed by the COVID-19 pandemic, governments can implement economic stimulus policies to create more job opportunities, reduce perceived employment pressures among college students, and alleviate job market uncertainties. Academic institutions can contribute by offering career guidance courses and providing career counseling services to enhance the students’ CDMS. These proactive measures can facilitate a smoother transition from the academic environment to the workforce for college graduates.

In the wake of the COVID-19 pandemic, employment trends have undergone significant shifts, which directly influence the job search experiences of college students. The pandemic has exacerbated economic uncertainty, with many industries facing disruptions and fluctuations in their hiring practices. For example, sectors such as tourism, hospitality, and retail have experienced severe job losses, while tech and healthcare industries have seen rapid growth. As a result, college students are navigating an increasingly competitive and volatile job market, where remote work, gig economy opportunities, and automation are becoming more prevalent. This post-pandemic landscape presents unique challenges for students, particularly those with limited work experience or skills that may not align with the demands of evolving industries. The heightened employment pressure faced by students, coupled with the lack of coping strategies to navigate these changes, underscores the importance of targeted CRSS. Academic institutions and career services need to adapt by offering resources that address these new realities.

### 4.2. Limitations and Future Directions

This research has some limitations. Firstly, regarding research methodology, this research is cross-sectional, and causal relationships among variables need to be interpreted with caution. Future research could employ longitudinal studies to validate our findings. Secondly, the data were solely based on self-reports from college students, and subsequent research could benefit from using data from different sources (e.g., parents, employers) for a more comprehensive analysis. Additionally, further distinctions could be made between the various types of CRSS and employment pressure, as different types of support and pressure may exert unique effects on the JSB. Lastly, this study’s data were exclusively drawn from Chinese mainland final-year college students and collected during COVID-19. Given variations in pandemic control policies across different regions and countries, as well as the differences in the economic impact of the pandemic, the employment environment may differ. Consequently, the conclusions of the study should be applied with caution to other age groups or countries.

## 5. Conclusions

The aim of this research is to investigate the underlying mechanisms of CRSS on JSBs among Chinese university students, and to test the indirect effect of CDMS and the moderating effect of employment pressure. Our research findings indicate that CRSS not only directly and positively predicts JSB in university students but also indirectly influences JSB through CDMS. Employment pressure positively moderated this direct relationship, such that when employment pressure was higher, the effect of CRSS on JSB was stronger. Additionally, employment pressure negatively moderated the indirect relationship between CRSS and JSB, such that when employment pressure was higher, the effect of CRSS on JSB via CDMS was weaker. These results shed light on how and when CRSS can promote JSBs among college graduates, further supporting the SCCT, extending the research literature, and providing valuable insights into the facilitation of a successful transition of college graduates from school to work. Based on these findings, career counselors should prioritize providing tailored CRSS interventions, particularly for students facing high employment pressure. This could include personalized career coaching, job search strategies, and skill-building workshops aimed at boosting self-efficacy. For policymakers, the findings suggest that creating a supportive job market environment through economic policies that reduce employment pressure is crucial. Initiatives such as job creation programs, internships, and partnerships between universities and industries could alleviate some of the stress students face as they transition into the workforce.

## Figures and Tables

**Figure 1 behavsci-15-00260-f001:**
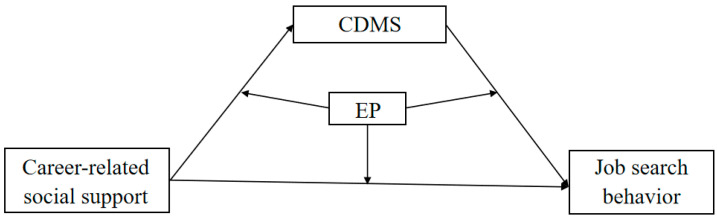
The hypothesized model. Note. CDMS = career decision-making self-efficacy, EP = employment pressure.

**Figure 2 behavsci-15-00260-f002:**
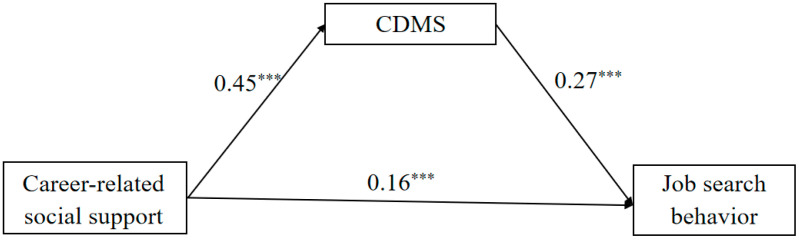
The mediating role of career decision-making self-efficacy. Note. CDMS = career decision-making self-efficacy. Note. *** *p* < 0.001.

**Figure 3 behavsci-15-00260-f003:**
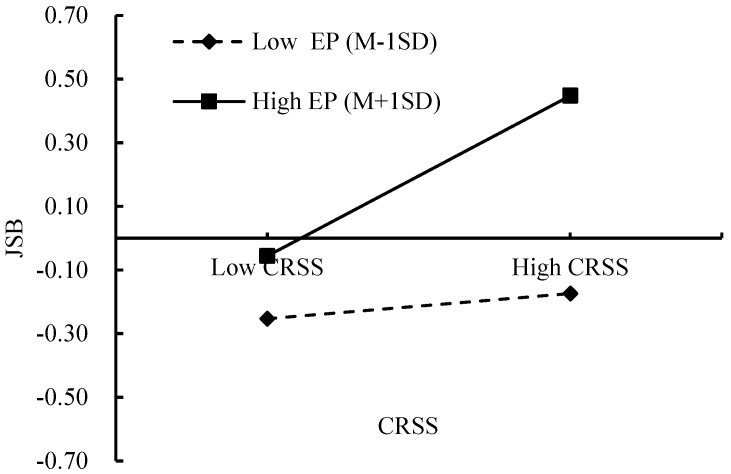
The interaction between EP and CRSS on JSB. Note. CRSS = career-related social support, EP = employment pressure, JSB = job search behavior.

**Figure 4 behavsci-15-00260-f004:**
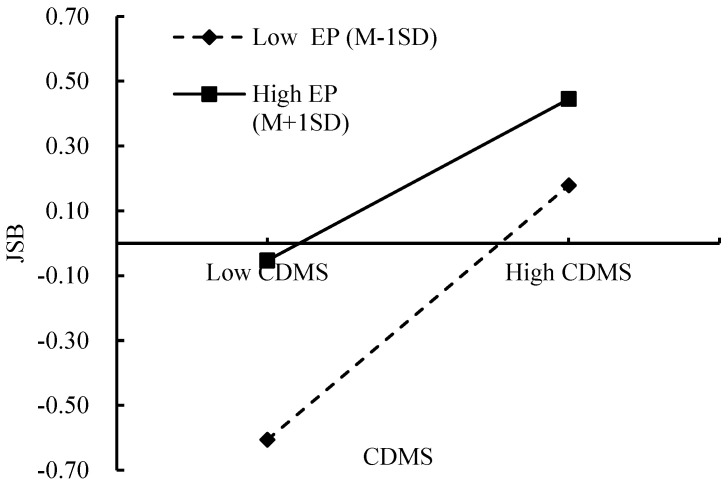
The interaction between EP and CDMS on JSB. Note. CDMS = career decision-making self-efficacy, EP = employment pressure, JSB = job search behavior.

**Table 1 behavsci-15-00260-t001:** The results of the descriptive and correlative analyses.

	M	*SD*	1	2	3	4
1. Career-related social support	3.38	0.97	1			
2. Career decision-making self-efficacy	3.54	0.59	0.45 **	1		
3. Employment pressure	3.22	0.81	–0.07	–0.22 **	1	
4. Job search behavior.	2.72	0.90	0.29 **	0.35 **	0.11 **	1

Note. ** *p* < 0.01.

**Table 2 behavsci-15-00260-t002:** Moderated Mediation Model Testing Results.

Outcome	Predictors	*R*	*R* ^2^	*F*	β	*t*
CDMS		0.51	0.26	35.01 ***		
	Gender				–0.01	– 0.31
	Hukou				0.09	1.17
	SES				0.20	3.92 ***
	EP				–0.19	–5.34
	CRSS				0.44	12.21 ***
	CRSS × EP				0.01	0.27
JSB		0.45	0.20	18.53 ***		
	Gender				–0.01	–0.28
	Hukou				0.24	2.89 *
	SES				–0.05	–1.01
	CRSS				0.15	3.49 ***
	CDMS				0.32	7.52 ***
	EP				0.20	5.22 ***
	CRSS × EP				0.11	3.01 **
	CDMS × EP				–0.07	–2.26 *

Note. * *p* < 0.05, ** *p* < 0.01, *** *p* < 0.001; Gender (0 = males, 1 = females), Hukou (0 = urban areas, 1 = rural areas); CRSS = career-related social support, CDMS = career decision-making self-efficacy, EP = employment pressure, JSB = job search behavior.

## Data Availability

Data can be available on reasonable request from the authors.

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
