# Peer review of "Linking Career-Related Social Support to Job Search Behavior Among College Students: A Moderated Mediation Model"

_behavsci, 2025, doi:10.3390/bs15030260_

Round 1
Reviewer 1 Report
Comments and Suggestions for Authors
This study contributes meaningfully to the understanding of how career-related social support (CRSS) influences job search behavior (JSB) among final-year college students, particularly during the COVID-19 pandemic. The use of Social Cognitive Career Theory (SCCT) provides a strong theoretical framework, and the study effectively identifies mediating and moderating variables that impact JSB.
While the results are well-explained, certain sections in the discussion could benefit from more explicit connections between theoretical implications and practical applications. For example, the employment pressure moderation effect could be further contextualized with real-world scenarios/examples.
While the paper is well-referenced, additional discussion of post-pandemic employment trends and their impact on student job searches would strengthen the paper’s relevance.
In the introduction, briefly mention alternative frameworks that have been used to study job search behavior and justify why SCCT is particularly suitable for this study.
In the methodology section, clarify the rationale for choosing the specific universities used in the sample.
In the conclusion, provide more concrete recommendations for career counselors and policymakers based on the findings.
Comments on the Quality of English LanguageSome sentences are overly complex and would benefit from clearer, more concise phrasing. A professional English language edit is recommended to improve clarity.
Author Response
Response to the reviewer 1 comments
This study contributes meaningfully to the understanding of how career-related social support (CRSS) influences job search behavior (JSB) among final-year college students, particularly during the COVID-19 pandemic. The use of Social Cognitive Career Theory (SCCT) provides a strong theoretical framework, and the study effectively identifies mediating and moderating variables that impact JSB.
Comments 1:While the results are well-explained, certain sections in the discussion could benefit from more explicit connections between theoretical implications and practical applications. For example, the employment pressure moderation effect could be further contextualized with real-world scenarios/examples.
Response:Thank you for your comments. We have expanded on the relevant content in the discussion section. For example, a graduate facing heightened employment pressure in a saturated job market may struggle to maintain motivation and resilience during the job search process. In such cases, tailored CRSS, such as personalized career counseling, skill-building workshops, or networking opportunities, can provide essential resources to manage these challenges. This support can help students regain their sense of control and optimism, mitigating the detrimental effects of employment pressure.
Comments 2:While the paper is well-referenced, additional discussion of post-pandemic employment trends and their impact on student job searches would strengthen the paper’s relevance.
Response:Thank you for your advice. We have added the relevant content in the discussion section in response to your suggestions. In the wake of the COVID-19 pandemic, employment trends have undergone significant shifts, which directly influence the job search experiences of college students. The pandemic has exacerbated economic uncertainty, with many industries facing disruptions and fluctuations in hiring practices. For example, sectors such as tourism, hospitality, and retail have experienced severe job losses, while tech and healthcare industries have seen rapid growth. As a result, college students are navigating an increasingly competitive and volatile job market, where remote work, gig economy opportunities, and automation are becoming more prevalent. This post-pandemic landscape presents unique challenges for students, particularly those with limited work experience or skills that may not align with the demands of evolving industries. The heightened employment pressure faced by students, coupled with a lack of coping strategies to navigate these changes, underscores the importance of targeted CRSS. Academic institutions and career services need to adapt by offering resources that address these new realities.
Comments 3:In the introduction, briefly mention alternative frameworks that have been used to study job search behavior and justify why SCCT is particularly suitable for this study.
Response:Thank you for your valuable feedback. We have incorporated the relevant content in the introduction section in response to your suggestions.
Several psychological frameworks, such as the theory of planned behavior (Song et al., 2006; Lin, 2010), expectancy–value theory (Manroop & Richardson, 2016), and self-regulation theory (van Hooft et al., 2021), have been widely used to study job search behavior. However, the social cognitive career theory (SCCT) is particularly suitable for this study as it emphasizes the dynamic interplay between cognitive, affective, and environmental factors in career-relevant activities (Lent & Brown, 2019).
Additional references:
van Hooft, E. A. J., Kammeyer-Mueller, J. D., Wanberg, C. R., Kanfer, R., & Basbug, G. (2021). Job search and employment success: A quantitative review and future research agenda. Journal of Applied Psychology, 106(5), 674–713. https://doi.org/10.1037/apl0000675
Manroop, L., & Richardson, J. (2016). Job search: A multidisciplinary review and research agenda. International Journal of Management Reviews, 18(2), 206–227. https://doi.org/10.1111/ijmr.12066
Lin, H. F. (2010). Applicability of the extended theory of planned behavior in predicting job seeker intentions to use job‐search websites. International Journal of Selection and Assessment, 18(1), 64–74. https://doi.org/10.1111/j.1468-2389.2010.00489.x
Song, Z., Wanberg, C., Niu, X., & Xie, Y. (2006). Action–state orientation and the theory of planned behavior: A study of job search in China. Journal of Vocational Behavior, 68(3), 490–503. https://doi.org/10.1016/j.jvb.2005.11.001
Comments 4:In the methodology section, clarify the rationale for choosing the specific universities used in the sample.
Response:Thank you for your suggestions. We have added the relevant content in the methodology section. The two universities are located in central China, near Wuhan, which is close to the epicenter of the COVID-19 pandemic, thereby providing a more accurate reflection of the real-world conditions during the pandemic.
Comments 5:In the conclusion, provide more concrete recommendations for career counselors and policymakers based on the findings.
Response:Thank you for your suggestions. We have added the relevant content in the conclusion section in response to your feedback.
Based on these findings, career counselors should prioritize providing tailored CRSS interventions, particularly for students facing high employment pressure. This could include personalized career coaching, job search strategies, and skill-building workshops aimed at boosting self-efficacy. For policymakers, the findings suggest that creating a supportive job market environment through economic policies that reduce employment pressure is crucial. Initiatives such as job creation programs, internships, and partnerships between universities and industries could alleviate some of the stress students face as they transition into the workforce.
Comments 6:Comments on the Quality of English Language
Some sentences are overly complex and would benefit from clearer, more concise phrasing. A professional English language edit is recommended to improve clarity.
Response:Thank you for your feedback. We have revised the sentences to make them clearer and more concise, ensuring improved clarity throughout the manuscript.
Reviewer 2 Report
Comments and Suggestions for Authors
The article presents the author's research results, based on which the key thesis was formulated that career-related social support (CRSS) significantly affects job search behavior (JSB). However, the mechanisms and boundary conditions through which CRSS affects JSB, especially during the COVID-19 pandemic, remain unclear. It is worth noting here that career-related social support (CSSR) refers to the type of support an individual receives from their environment - family, friends, mentors, colleagues, teachers or other people

Author Response
Response to the reviewer 2 comments
Comments and Suggestions for Authors
The article presents the author's research results, based on which the key thesis was formulated that career-related social support (CRSS) significantly affects job search behavior (JSB). However, the mechanisms and boundary conditions through which CRSS affects JSB, especially during the COVID-19 pandemic, remain unclear. It is worth noting here that career-related social support (CSSR) refers to the type of support an individual receives from their environment - family, friends, mentors, colleagues, teachers or other people.In a global perspective, research results can be compared across countries and cultures. This allows for the identification of universal adaptation mechanisms and understanding how different educational and social systems support students in times of crisis.
Comments 1: too extensive introduction. The use of first-level subchapters in the introduction is not understandable (example 1.1. Career-related social support and job search behavior ). This obvious weakness must be treated appropriately to the content of the article, e.g. as a stand-alone chapter.
Response: Thank you for your feedback. In response to your suggestion, we have removed the second-level subheadings in the introduction to enhance the readability and flow of the section, ensuring a more cohesive and continuous presentation of the content.
Comments 2:In the second chapter, using first-degree subchapters similarly to the introduction – this is a clear manner of the authors of the article – an introduction is necessary. The simplest solution in this situation would be to refrain from dividing the chapter into first-degree subchapters, while maintaining a homogeneous chapter.
Response: Thank you for your feedback. In response to your suggestion, we have removed the second-level subheadings in the introduction to enhance the readability and flow of the section, ensuring a more cohesive and continuous presentation of the content.
Comments 3:The excessively extensive content of the third chapter of the article in relation to the second and fourth chapters; Very modest content of the third chapter – Results. This chapter is the most important – due to the cognitive values and methodological innovation of the research – part of the article and it focuses on statistical proof of the correctness of the research. This is a serious omission. This chapter should be radically changed by shifting the focus from statistical proof to the research results and documenting their significance in the light of the formulated theses .
Response: Thank you for your advice. In line with the journal's standard practice, most researchers typically present statistical results in the Results section rather than interpreting them, to avoid overlap with the Discussion section. However, based on your feedback, we have made appropriate adjustments by adding relevant content in certain paragraphs to improve the readability of the article and better highlight the significance of the research findings.
Comments 4:Very modest conclusions and summary prove the low editorial culture of the article,
I am not convinced by the content of the thesis that the research results shed light on how and when CRSS can promote JSB among college graduates, additionally supporting SCCT and expanding the research literature and providing valuable insights to help college graduates successfully transition from school to work. The content lacks a conclusion - replacing the above simple thesis - indicating the need to continue and develop the research presented in the article. In the reviewer's opinion, it should be to strongly advance and expand research on how and when career-related social support (CRSS) can promote job search behavior (JSB) among college graduates. The following conclusion provides the following argumentation to support this claim, while also highlighting the benefits for social cognitive career theory (SCCT), the research literature, and professional practice. In conclusion, it seems that improving the article by compensating the critical remarks contained in the review would significantly increase its cognitive and editorial value.
Response: Thank you for your valuable suggestions and expanded perspective. In response to your feedback, we have revised and enhanced both the Discussion and Conclusion sections to better address the significance of the research results and to provide a stronger argument for the need to further develop research on how and when Career-Related Social Support (CRSS) can promote Job Search Behavior (JSB) among college graduates. The specific content added is as follows; please refer to the manuscript for the full context and details.
This study, grounded in the SCCT, examined the mechanisms and boundary conditions through which CRSS influences JSB in college students. Specifically, the research highlights the critical role of CRSS in shaping JSB, particularly during challenging times such as the COVID-19 pandemic or periods of high employment pressure. By examining how CRSS not only directly impacts job search efforts but also acts as a buffer in times of stress, this study provides valuable insights into the ways social support can mitigate the negative effects of employment pressure on students' career development. Furthermore, this research identifies key boundary conditions, such as the moderating effect of employment pressure, that influence the efficacy of CRSS in promoting job search behavior. These findings underscore the importance of considering external contextual factors when designing interventions to support students. In response to existing research gaps, the study proposes the design of tailored mentoring programs that integrate CRSS and employment pressure management strategies. Such programs can provide personalized support, enhancing students' coping mechanisms and job search strategies. Finally, the findings have important implications for public policy. Policymakers can use this research to inform the development of policies that create supportive environments for college graduates from academia to the workforce.
In the wake of the COVID-19 pandemic, employment trends have undergone significant shifts, which directly influence the job search experiences of college students. The pandemic has exacerbated economic uncertainty, with many industries facing disruptions and fluctuations in hiring practices. For example, sectors such as tourism, hospitality, and retail have experienced severe job losses, while tech and healthcare industries have seen rapid growth. As a result, college students are navigating an increasingly competitive and volatile job market, where remote work, gig economy opportunities, and automation are becoming more prevalent. This post-pandemic landscape presents unique challenges for students, particularly those with limited work experience or skills that may not align with the demands of evolving industries. The heightened employment pressure faced by students, coupled with a lack of coping strategies to navigate these changes, underscores the importance of targeted CRSS. Academic institutions and career services need to adapt by offering resources that address these new realities.
Based on these findings, career counselors should prioritize providing tailored CRSS interventions, particularly for students facing high employment pressure. This could include personalized career coaching, job search strategies, and skill-building workshops aimed at boosting self-efficacy. For policymakers, the findings suggest that creating a supportive job market environment through economic policies that reduce employment pressure is crucial. Initiatives such as job creation programs, internships, and partnerships between universities and industries could alleviate some of the stress students face as they transition into the workforce.
Round 2
Reviewer 2 Report
Comments and Suggestions for Authors
I propose to include the article in the publishing plans of Behavioral Sciences (ISSN 2076-328X)